

# Modelling rock wall permafrost degradation in the Mont Blanc massif from the LIA to the end of the 21[st] century

Florence Magnin[1], Jean-Yves Josnin[1], Ludovic Ravanel[1], Julien Pergaud[2], Benjamin Pohl[2], Philip Deline[1]

[1] EDYTEM Lab, Université Savoie Mont Blanc, CNRS, 73376 Le Bourget du Lac, France
[2] Centre de Recherches de Climatologie, Biogéosciences, Université de Bourgogne Franche-Comté, CNRS, Dijon, France

*Correspondence to*: Florence Magnin (florence.magnin@univ-smb.fr)

**Abstract.** High alpine rock wall permafrost is extremely sensitive to climate change. Its degradation can trigger rock falls constituting an increasing threat to socio-economical activities of highly frequented areas. Understanding of permafrost evolution is therefore crucial. This study investigates the long-term evolution of permafrost in three vertical cross-sections of rock wall sites between 3160 and 4300 m a.s.l. in the Mont Blanc massif, since LIA steady-state conditions to 2100. Simulations are forced with air temperature time series, including two contrasted air temperature scenarios for the 21[st]
century representing possible lower and upper boundaries of future climate change according to the most recent models and climate change scenarios. The model outputs for the current period (2010-2015) are evaluated against borehole temperature measurements and an electrical resistivity transect: permafrost conditions are remarkably well represented. Along the past two decades, permafrost has disappeared into the S-exposed faces up to 3300 m a.s.l., and possibly higher. Warm permafrost (*i.e.* > -2°C) has extended up to 3300 and 3850 m a.s.l. in N and S-exposed faces, respectively. Along the 21[st] century, warm
permafrost is likely to extent at least up to 4300 m a.s.l. into the S-exposed rock walls, and up to 3850 m a.s.l. at depth of the N-exposed faces. In the most pessimistic case, permafrost will disappear at depth of the S-exposed rock walls up to 4300 m a.s.l., whereas warm permafrost will extend at depth of the N faces up to 3850 m a.s.l., but could disappear at such elevation under the influence of a close S face. The results are site-specific and extrapolation to other sites is limited by the imbrication of the local topographical and transient effects. Shorter time-scale changes are not debatable due to limitations in the
modelling approaches and future air temperature scenarios.

**Keywords:** rock wall, permafrost degradation, Mont Blanc massif, global warming, bi-dimensional modelling

## 1 Introduction

The IPCC Fifth Assessment Report (AR5) draws a global increase in permafrost temperature since the 1980s (IPCC, 2014).
By the end of the 21[st] century, the near-surface permafrost area is projected to retreat by 37 to 81% according to RCP 2.6 (Representation Concentration Pathways with a projected increase in radiative forcing of 2.6 W.m[-²]; Vuuren et al., 2011) and RCP 8.5, respectively. Concerns about natural disasters resulting from mountain permafrost degradation have started to rise





during the late 1990s (IPCC, 1996). Haeberli et al. (1997) identified various types of high mountain slope instabilities prepared or triggered by interactive processes between bedrock, permafrost and glaciers. Although the examples were scarce, this warning study has been largely confirmed during the past two decades, especially with the increase in rock fall activity of high-elevated permafrost rock walls (Ravanel and Deline, 2011).

Since the hot summer of 2003 and the remarkable number of rock falls observed in the European Alps (Schiermeier, 2003; Ravanel et al., 2011), rock wall permafrost has been intensively studied in various mountain areas (Gruber, 2005; Noetzli, 2008; Allen et al., 2009; Hasler, 2011; Hipp, 2012; Magnin, 2015). The role of permafrost degradation in rock wall stability is more and more admitted (*e.g.*, Krautblatter et al., 2013) and mountain permafrost is of high concern for construction practices (Harris et al., 2001a; Bommer et al., 2010). The destabilisation of rock wall permafrost endangers high mountains
activities, infrastructures (Duvillard et al., 2015), mountain-climbers and workers. Valley floors could be affected by high mountain hazards owing to the possible cascading effects (Deline, 2001; Einhorn et al., 2015).

Rock wall permafrost is highly sensitive to climate change because (i) it is directly coupled with the atmosphere (absence of debris and seasonal snow cover), (ii) the delaying effect of latent heat processes is reduced due to the low ice content (Smith and Riseborough, 1996), and (iii) it is subject to multi-directional warming from the different summit sides (Noetzli et al.,
2007). Therefore, it is prone to much faster changes than any other kind of permafrost (Haeberli et al., 2010).

The monitoring of rock wall permafrost has started in the late 1990s in Switzerland with the drilling of two boreholes at the Jungfraujoch site (PERMOS, 2004). A latitudinal transect along European mountains has been later installed in the framework of the PACE project (Sollid et al., 2000; Harris et al., 2001b; Harris et al., 2009). A warming trend clearly appeared over the past decade in most of the existing boreholes (Blunden and Arndt, 2014).

The presence of ice in the fractures of steep alpine bedrock has been demonstrated by engineering work (Keusen and Haeberli, 1983; King, 1996; Gruber and Haeberli, 2007). This ice highly contributes to rock wall stability because it increases the tensile and shear strengths of the fractures (Davies et al., 2001; Krautblatter et al., 2013). The warming of an ice-filled fracture has two effects on its stability: the loss of bonding and the release of water which increases the hydrostatic pressure. An ice-filled fracture becomes critically unstable by between -1.4 and 0°C (Davies et al., 2001). In this way, the
warming of permafrost and the thickening of the active layer by heat conduction could be responsible for rock wall destabilization (Gruber and Haeberli, 2007). But heat advection through the circulation of water supplied by the melting of the interstitial ice, snow or glacier ice could warm permafrost at deeper layers than those reached by heat conduction (Hasler et al., 2011a). Hydraulic and hydrostatic pressures in frozen bedrock are modified under freezing and thawing, and can be involved in rock wall destabilisation throughout a large range of processes (for a review of these processes see Matsuoka and
Murton, 2008; Krautblatter et al., 2012).



Historical and recent rock fall events have been systematically inventoried in the Mont Blanc massif (Ravanel et al., 2010a; Ravanel and Deline, 2013). Their trend revealed a clear relationship with hot climate signals at various time scales from seasonal to decadal (Ravanel et al., 2010b; Ravanel and Deline, 2011; Huggel et al., 2012). Given the recent evidences, one can assert that the magnitude and frequency of these hazards are likely to increase along the 21[st] century of projected global warming (IPCC, 2011). Knowledge about the current and future thermal state of the Mont Blanc massif rock walls is thus required in order to take into account a risk that threatens activities in this densely frequented high mountain area.

Patterns and processes of long-term permafrost changes in steep mountain flanks were studied in idealized cases for the European Alps (Noetzli et al., 2007; Noetzli and Gruber, 2009) and Norway (Myhra et al., 2015). But future changes in rock wall permafrost driven by the most recently released RCPs have not been addressed yet, whereas the site-specific response to 21[st] century climate change has not been considered. Furthermore, evaluation of time-dependent rock wall permafrost models has remained limited by the lack of empirical data. To address site-specific long-term changes in rock wall permafrost of the Mont Blanc massif we run 2D models on NW-SE cross-sections of three sites covering an elevation transect, starting from 3160 up to 4300 m a.s.l., which encompasses currently warm and cold permafrost conditions. Transient simulations are run from the end of the LIA (1850 CE) to the end of the 21[st] century (2100) based on two different RCPs (4.5 and 8.5) accounting for moderately optimistic and pessimistic scenarios. Bi-dimensional models of the current period (2010-2015) are benchmarked against an independent data set in order to evaluate the model performance. Underlying research questions are the following:

- Is our modelling approach suitable to reproduce current permafrost conditions at the site scale?

- How permafrost has changed within these sites along the past decades?

- How rock wall permafrost will possibly evolve by the end of the 21[st] century considering the latest IPCC projections?

This study provides insight as much in the recent changes of rock wall temperature as in its future evolution, usable for retrospective analyses of rock wall instability and for assessing future hazards.

## 2 Study site and available data

The Mont Blanc massif is an external variscan high mountain range culminating at 4809 m a.s.l., located on the western margin of the European Alps (Fig. 1). Its two major lithological units are a polymetamorphic basement along its western margin, and a unit of Mont Blanc granite at its core (Bussy and von Raumer, 1994). It covers c. 550 km² over France, Switzerland and Italy, of which c. 30 % are glaciated (Gardent et al., 2014; Fig. 1). About 65 % of its rock walls above 2300 m a.s.l. are permanently frozen, according to a first estimation of permafrost distribution on the French side and borders (Magnin et al., 2015a; Fig. 1). For the purpose of this study, we selected three sites at various elevations and under various



permafrost conditions: Aiguille du Midi, Grands Montets, and Grand Pilier d'Angle. All the three sites are located in the granitic area of the massif. Their elevations as well as their permafrost conditions are representative of the Mont Blanc massif rock walls.

### 2.1. Aiguille du Midi and bedrock temperature data

Studies on rock wall permafrost have started by the end of 2005 in the Mont Blanc massif with the steady installation of 9 rock surface temperature (RST) sensors at the Aiguille du Midi summit (AdM), a set of three granite pillars. The ADM is accessible by cable-car throughout the year (500,000 visitors per year). As a pilot site in high-elevated permafrost research, the AdM is now equipped with a variety of instruments to measure rock wall temperature (Magnin et al., 2015b), snow cover (Magnin et al., submitted) and mechanics (Ravanel et al., 2016). Three 10-m-deep boreholes of 15-nodes-thermistor chains

are installed in the AdM bedrock and register temperature with a 3h time step since December 2009 (NW and SE faces) and April 2010 (NE face). Therefore, the AdM has been chosen because of the possibility to quantitatively evaluate the model outputs. It is characterized by the coexistence of cold permafrost (*c.* -4.5°C at 10-m-depth) on its NW face and warm permafrost (*c.* -1.5°C at 10 m depth) on its SE face (Fig. 2). Thermal effects of snow are observed in the three boreholes (Magnin et al., 2015a). The local cooling effect of a fracture has been detected at 2.5 m depth of the NW borehole.

Nevertheless, temperature at 10-m-depth seems mainly governed by conductive heat transfer processes and lateral heat fluxes from the warm South face to the cold North face.

### 2.2. Grands Montets and ERT data

The Grands Montets (GM) is a summit culminating at 3296 m a.s.l., lying northbound and *c.* 800 m below the Aiguille Verte (4122 m a.s.l.). In 1962-63 a cable car was installed on its mid-steep North face (*c.* 60°) to transport skiers up to the glaciated

area. In May 2011, a RST logger was installed (GEOPrecision PT1000, sensor accuracy ±0.1°C) at the foot of the highly fractured NW face (3058 m a.s.l.) in a 85° steep rock wall portion. It recorded the rock temperature at depths of 3, 10, 30 and 55 cm until January 2013. The 2012 mean annual rock surface temperature (MARST) at a depth of 3 cm was -1.4°C. In 2012 and 2013, electrical resistivity tomography (ERT) soundings have been conducted along the NW face of the GM and 4 other sites of the massif (Magnin et al., 2015c). The potential of ERT for qualitative evaluation of 2D permafrost models has been

demonstrated by Noetzli et al. (2008). ERT covers a much wider and deeper rock wall portion than bedrock temperature. In a way, that makes ERT a better approach to evaluate distributed models of rock wall permafrost because it has the capacity to represent the spatial variability of rock wall permafrost (Magnin et al., 2015c). Conversely, direct temperature measurements allow for quantitative evaluation, but have the disadvantage to be only representative for the measurement point. We selected the GM site as a second site because (i) a 160-m-long and 25-m-deep ERT transect is available for model evaluation, (ii) the

site bears socio-economical interests with *c.* 200 000 persons using the cable car every year, and (iii) it is located within the warm permafrost fringe of the massif as revealed by the RST data, permafrost map (Fig. 1 and 2) and the ERT transect.



Moreover, this site has been regularly affected by rock falls during the last decade which supports the interest in studying its thermal dynamics.

## 2.2. Grand Pilier d'Angle

Finally, we chose the third site based on its elevation in order to also consider an entirely cold permafrost site. We chose the
Grand Pilier d'Angle (GPA, 4304 m a.s.l.) where cold permafrost is likely to be present on all the rock faces according to the permafrost map (Fig. 2). The East face of the GPA was strongly affected by a rock avalanche in November 1920. About 3 millions m$^3$ of rock detached from the face in several stages and travelled onto the Brenva Glacier on a distance > 5 km, reaching the valley floor (Deline et al., 2015). Because of its altitude, its height (900 m), its stiffness (rock walls often subvertical) and its remoteness, the GPA includes climbing routes among the most difficult and exposed of the Mont Blanc
massif. For this last site located on the Italian side of the massif, no data set is available for model evaluation, and the quality of the 2D models will be assessed based on the evaluation of the two other sites.

## 3 Modelling approach

### 3.1. Background and strategy

Rock wall permafrost is a relatively simple system since it has no debris or spatially and temporally continuous snow covers
such as on gentler mountain slopes: it is straightforwardly coupled with the atmosphere. Therefore, it is mainly governed by air temperature and incoming solar radiations (Gruber et al., 2004), whereas patchy and intermittent snow deposits could further cool the bedrock (Hasler et al., 2011b). At depth, the temperature in hard rock mainly depends on the conductive heat transfer from the surface (Williams and Smith, 1989; Wegmann et al., 1998), and 3D heat fluxes induced by the aspect-dependent RST variability (Noetzli et al., 2007).
Generally, modelling procedures of permafrost rock wall first calculate the RST and then solve the heat conduction equation to simulate subsurface temperature. Pioneer studies used distributed energy balance models to calculate the RST (Gruber et al., 2004) and simulated the subsurface temperature fields with the mere consideration of (i) conductive heat transfer within idealized high mountain geometry and (ii) latent heat processes to account for water phase changes in the bedrock interstices (Noetzli et al., 2007; Noetzli and Gruber, 2009).
Due to the high computational efforts in energy balance approaches, statistical approaches were later adopted to compute the RST (Allen et al., 2009; Hipp et al., 2014; Myhra et al., 2015). The increasing amount of available RST time series in the European Alps has permitted the formulation of such statistical model for the entire Alpine range (Boeckli et al., 2012). This last model has been applied on a 4-m-resolution DEM of the French part and Italian border of the Mont Blanc massif with local air temperature input data to map the MARST (Fig. 1 and 2, Magnin et al., 2015a).
In our modelling procedure, we use the MARST map available for the French part of the Mont Blanc massif to generate the initial RST condition. We run the transient simulations in the commercial software *Feflow* (DHI-WASY) by forcing the RST



with climate time series from 1850 to 2100, and solving the heat conduction equation in 2D with consideration of freeze and thaw processes in the bedrock interstices.

### 3.2. Boundary conditions

### 3.2.1. Rock surface temperature

We first extracted the topography and the MARST from the 4 m resolution DEM (provided by RGD 73-74), and the MARST mapped over it to serve as upper boundary conditions along NW-SE vertical transects (Fig. 2). The MARST map has been evaluated against 43 measurement points of RST from the multi-year time series of the 9 RST loggers installed around the AdM. The modelled MARST tend to underestimate measured MARST values of sun-exposed rock faces and to over-estimate those of the shaded faces. Nevertheless, the mean bias of -0.21°C (Magnin et al., 2015a) indicates a generally

good approximation of the real-world MARST at this site.

The linear regression used to produce the MARST map has been formulated with the mean air temperature of the 1961-1990 reference period (homogenised by Hiebl et al., 2009), and measured MARST adjusted to the reference period. The MARST were adjusted by applying the difference in air temperature between the reference period and the years of the MARST measurements (Boeckli et al., 2012). In our modelling procedure, we considered that the MARST extracted from the map is

representative of the year 1961. Then, we lowered this MARST by 1°C to approximate the MARST at the end of the LIA (Auer et al., 2007; Böhm et al., 2010) and set up the initial RST condition at the upper boundary of the model domain (Fig. 3). MARST differences driven by topo-climatic factors clearly appear along the extracted profiles but are site specific. At the GM, the MARST difference between the SE and NW face is only *c.* 1°C. It is of *c.* 5°C at the AdM and *c.* 6°C at the GPA. These variable temperature differences for similar aspect differences (180°) are attributed to two factors: the differences in

slope steepness and the local shading. The GM and AdM are isolated summits with no close shading. Conversely, the GPA NW face is located right below the Mont Blanc which shades the GPA West-exposed faces and lowers their MARST. The mid-steep NW slope and rounded summit of the GM receive solar radiations a larger part of the daylight than in sub-vertical settings and with a more perpendicular incidence of the beams.

Starting from the initial RST representative of the LIA conditions, we first initialise 2D steady-state temperature fields for

the year 1850, and then run transient simulations using reconstructed, measured and projected climate time series until 2100 (see Sect. 3.3).

### 3.2.2. Model geometries

Below the topographical profiles, a box of a height of 5000 m was added to shut off the model geometry of each site. A constant geothermal heat flux of 85 mW.m⁻² (Medici and Rybach, 1995; Maréchal et al., 2002) was set up as lower boundary

condition. Above these boxes, a finite element mesh with triangular elements was generated to discretise the subsurface material (Fig. 3). Even though the spatial resolution of the boundary conditions is 4 m, we refined the meshes close to the



surface in order to catch up the near surface temperature gradient. The spatial resolution of the initial RST, based on the 4 m resolution map, was refined accordingly using linear interpolation. This mesh and RST refinement does not provide much information at depth, nor improve the quality of the models, but facilitates the model evaluation. At greater depth, we kept a mesh size of 4 m, in coherence with the resolution of the input data.

This approach resulted in a 8548 nodes and 16141 elements MESH at the AdM, 5844 nodes and 10952 elements at the GM, and 12087 nodes and 23344 elements at the GPA (Fig. 3).

On the AdM site, 37 observation points were defined between the surface and 10 m depth of the NW and SE faces at the location of the boreholes (Sect. 2.1). The mesh was refined along these observations points (Fig. 3) and simulated bedrock temperature is extracted at user defined time step during the transient simulations to serve for model evaluation (Sect. 4.2.1).

**3.3. Transient simulations**

**3.3.1. Initial condition**

To define an initial 2D temperature field in the model geometries, we run the heat transfers with the upper boundary condition (the 1850 RST) and lower boundary condition (the geothermal heat flux) until we reach steady-state conditions, balancing the respective influence of the atmosphere and geothermal heat fluxes. Models run for 80 000 ± 10 000 years

before achieving steady-state conditions. After this initialisation procedure that provides an initial condition for 1850, we run transient simulations with air temperature forcing time series from 1850 to 2100.

**3.3.2. Forcing data**

The transient simulations are forced with air temperature time series created from various sources of data. The temporal resolution of these forcing data was gradually refined with the increasing quality of the available data and the periods of

interest (Fig. 4). Along the period 1850-1961, no continuous air temperature measurements are available for the Mont Blanc massif. Therefore, we assumed a linear increase of 1°C between 1850 and 1961 (Auer et al., 2007; Böhm et al., 2010), and run the simulations at an annual time step. A sensitivity analysis to higher time step did not change the final results for the periods of interest.

For the period 1961-1993, a climate time series was created at a monthly time step in the scope of the MARST mapping

based on measured temperature values in Chamonix (Magnin et al., 2015a). We used this monthly time series to force the model between 1961 and 1993, which constituted a first step in temporal resolution refinement of the forcing data.

In 1993, *Météo France* started continuous records of air temperature at hourly time step. From these hourly records, daily air temperature can be reliably calculated. Therefore, we forced the transient models with this daily air temperature time series between 1993 and 2015.

Finally, two contrasted scenarios were retained for the 21[st] century. Time series consist in daily 2 m air temperature simulated by the IPSL-CM5A-MR Earth System Model (Dufresne et al., 2013) that participated to the 5[th] Coupled Model




Inter-comparison Project (CMIP5, Taylor et al., 2012) / AR5 of IPCC (2013). For this study and for climate projections in the future decades we used two contrasted radiative forcing scenarios, namely the Representative Concentration Pathway (Moss et al., 2008) RCP4.5 and RCP8.5. They respectively correspond to an increase of +4.5 W.m$^{-2}$ and +8.5 W.m$^{-2}$ for 2100 relative to pre-industrial values, resulting in an air temperature increase of respectively +3 and +5°C by the end of the 21[st]

century according to the comparison of the measured mean air temperature of the 1980-2010 period and the projected mean air temperature for the period 2070-2100.

For RCP4.5 anthropogenic greenhouse gases emissions peak around 2040 and next decline (moderately optimistic scenario), while for RCP8.5 emissions continuously increase throughout the century (pessimistic scenario). The IPSL-CM5A-MR model was chosen because (i) its spatial resolution is among the highest among the CMIP5 model (1.25 × 2.5°), allowing for

more realistic orographic effects in the simulated climate; (ii) its basic state is very close to the recent observational records during the first years of the 21[st] century (Fig. 4); and (iii) its response to the radiative forcing throughout the century is close to the median of the CMIP5 models, ensuring a representative behaviour to estimate long-term evolutions. Analysed time series are obtained by extracting the closest grid-point (1336 m a.s.l) to the Mont Blanc Massif.

### 3.3.3. Heat transfers: conceptual approach

Rock wall permafrost is composed of rock, ice and air in non-saturated conditions. Sass (2003) has approached alternating saturated/unsaturated conditions under freezing and thawing of the near-surface pore spaces of a rock wall by mean of geophysical soundings. But the rates of saturation of alpine rock walls remain fairly misunderstood. Thus, the numerical models of rock wall permafrost have so far considered a saturated, homogeneous and isotropic media (Wegmann et al., 1998; Noetzli et al., 2007; Hipp et al., 2014; Myrha et al., 2015). Nonetheless, such approaches have been shown satisfactory

to simulate long-term temperature changes in alpine rock masses. Shorter time-scale processes are clearly a hydrogeological problem.

In the scope of this study, we used the hydrogeological software *Feflow* version 7.0 combined with the plug-in *Pi-Freeze* 1.0 which accounts for freeze and thaw processes. As a very first use, we adopted existing approaches of long term simulations (saturated, homogeneous and isotropic media) to simulate transient thermal processes along centennial time-scales. Further

developments will use the potential of *Feflow* to simulate with various saturation rates and fluid transfers.

### 3.3.4. Heat transfers: numerical approach

The conservation equation of energy for advective dispersive-diffusive transport of thermal energy depends on fluid flows (in saturated or unsaturated conditions, *i.e.* Darcy law incorporated in continuity equation or Richards equation), but works in pure conduction when the flows are zero. It is usually expressed as follows (Diersch, 2002):

$$\frac{\partial\left[\left(\varphi\rho_C\right)_L + (1-\varphi)\left(\rho_C\right)_S\right]T}{\partial t} = -\nabla.\left[\left(\rho_C\right)_L qT - \Lambda\nabla T\right]$$
(1)



with $\varphi$ the porosity (a-dimensional), $\rho_{CL}$ and $\rho_{CS}$ the volumetric heat capacities (J.m$^{-3}$.K$^{-1}$) of the liquid and solid phases respectively, $\Lambda$ the hydrodynamic thermal dispersion tensor, (J.m$^{-1}$.s$^{-1}$.K$^{-1}$) that includes thermal conductivity, $T$ the temperature (K) and $q$ the apparent flow velocity from Darcy or Richards equation (m.s$^{-1}$), to adjust in order to include both the air and the ice phases. The ice is included in the solid phase in order to modify only one parameter of thermal conductivity (and not the one related to fluids). The solid thermal conductivity $\lambda$ (W. m$^{-1}$. K$^{-1}$) remains:

$$\lambda_s = \lambda_{s,0} + \frac{\varepsilon_i \left( \lambda_i - \lambda_s \right)}{1 - \varepsilon} \tag{2}$$

with $\varepsilon_s$ the bulk fraction of the solid (rocks) $\varepsilon_i$ the bulk fraction of the ice and $\varepsilon$ the bulk fraction occupied both by water and air (Clausnitzer and Mirnyy, 2015). In the solid, the thermo-dispersion tensor is linked to the thermal conductivity through the solid bulk volume fraction, and is sufficient here, the fluid convection being out of the purpose of the present paper. Concerning the addition of the ice in the whole modelled medium, it is expressed throughout the bulk volume as: $\varepsilon_a + \varepsilon_w + \varepsilon_i + \varepsilon_r = 1$, with $\varepsilon_a$ the bulk fraction of air, $\varepsilon_w$ the bulk fraction of water, $\varepsilon_i$ the bulk fraction of ice and $\varepsilon_r$ the bulk fraction of rock. A relation is established between ice and liquid: called the mass fraction per bulk volume of the unfrozen liquid to the total liquid mass or freezing function:

$$F = \frac{\varepsilon_w \rho_w}{\varepsilon_w \rho_w + \varepsilon_i \rho_i} \tag{3}$$

where $\rho$ is the density of the corresponding phase ($_i$ for ice and $_w$ for water). This function F decreases with the fraction of ice. When the freezing point is $T_0$, then the ice forms gradually within a predefined temperature interval $\left[ T_0 - \frac{\Delta T}{2} , T_0 + \frac{\Delta T}{2} \right]$ of the length $\Delta T$.

### 3.3.5. Thermal parameters

The thermal conductivity of the rock was set to 3 W.m$^{-1}$.K$^{-1}$ which stands for a conservative value for saturated granitic rock (Cho et al., 2009). However, the thermal conductivity of a saturated media doesn't only depends on the mineral properties, but also on the water state, the ice being up to six times more conductive than the water at 0°C (Williams and Smith, 1989). Thermal conductivity variations along freeze and thaw cycles are accounted for by *Pi-Freeze* (see Eq. 2). The heat capacity of the rock was set to 1.8 MJ.m$^3$.K$^{-1}$.

In addition to the usual adjustable parameters of *Feflow*, *Pi-Freeze* allows a user-defined freezing temperature, a customisable temperature interval for freeze/thaw combined with a linear freezing function, adaptable thermal properties of ice, a user-specified residual fluid content and a configurable latent heat. Such possibilities are highly promising for adapting the modelling approach to the natural conditions.




To account for the latent heat processes related to the freeze and thaw of the interstitial ice contained in pores and fractures, we took a 5 % porosity value following the procedure from Noetzli et al. (2007). This value is just on the top acceptable values for dense crystalline rocks (Domenico and Schwartz, 1997) or lowly fissured crystalline rocks (Banton and Bangoy, 1999). Indeed, dense crystalline rock porosity without any fissure is usually inferior to 1 % while fractured crystalline rock

porosity quickly reaches values greater than 5 %. The 5 % value chosen here accounts then for the ice contained in fractures since bedrock discontinuities are not included.

Water contained in artificial pore-spaces is subject to a supercooling, *i.e.* a temperature deviation from the equilibrium freezing point at 0°C, until it reaches a spontaneous freezing point which depends on pore size and material (Alba-Simionesco et al., 2006). Geophysical experiments on various hard rock samples and under controlled laboratory settings

have quantified a freezing point depression of -1.2°C ±0.2°C (Krautblatter, 2009) due to the pressure and water salinity. To account for this supercooling characteristic of interstitial water, temperature freezing point $T_0$ (see 3.3.4) was set to -1°C in the *Feflow Pi-Freeze* module, while the temperature interval $\Delta T$ of the freezing function was set to 1°C.

The latent heat of fusion was set to 334 kJ. kg$^{-1}$.

## 4 Results

We here present the results of the simulations in three steps. First, we describe the permafrost conditions and changes between the steady-state at the end of the LIA to time-dependent conditions during the recent period. The recent conditions are illustrated through model snapshots in September 1992 and September 2015, displaying the active layer patterns in the uppermost layers; they are not presented since they go beyond the scope of this study and lie in the limits of our modelling approach (Sect. 5). In a second step, the model outputs for the recent period (2010-2015) are compared to an independent

data set of real world conditions for assessing the quality of the simulations along the 20[th] and early 21[st] centuries. Finally, after model evaluation, thermal conditions by the end of the 21[st] century in response to RCP4.5 and 8.5 forcings are presented.

### 4.1. Permafrost evolution from the LIA to the current period

#### 4.1.1. Steady-state at the LIA termination

Equilibrium conditions for the end of the LIA (1850) are displayed in Figure 5a for the three sites. In 1850, the GPA and the AdM were totally in cold permafrost (< -2°C). At the GM, a cold permafrost body was characterizing the NW subsurface and was extending below the SE face between *c.* 3260 and 3280 m a.s.l. Warm permafrost was already occupying most of the site, including the top.

The shape of the isotherms varies from one site to another, depending on the topographical settings. The steepest site (AdM)

shows sub-vertical isotherms down to *c.* 3720 m a.s.l., where they incline downwards and towards the NW to become more oblique. In the top part of the less steep site (GM), the -2°C isotherm is rather oblique whilst the -1°C one is sub-horizontal





in the lower part. In the GPA, isotherms are vertical in the top part and oblique in the middle part. In the lower part of the SE face, in the > -5°C area, isotherms obliquity decline to become more parallel to the upwards geothermal heat flow.

The modelled temperature gradients directly depend on the temperature difference between NW and SE flanks (Sect. 3.2.1): small temperature gradients are visible in the GM cross section, in accordance with the initial RST difference, whereas they

are higher in the sharper two other sites with more contrasted RST.

### 4.1.2. Transient temperature fields along the 20th and early 21st centuries

Figure 5b displays time-dependent conditions for the early 1990s, while Figure 5c exhibits those in 2015 after the two past decades of strong air temperature increase (Fig. 4). Along the 20th and early 21st centuries, permafrost has degraded in all the three sites. Warm permafrost has extended in the entire GM site and started to penetrate below the AdM SE face.

But this degradation shows site-specific features in terms of isotherm shapes and temperature field distributions.

At the GM, the cold permafrost body has subsisted until the early 1990s below the NW face, but has narrowed down to two small bodies. Meanwhile, the lower limit of the -1°C isotherm has risen up below the SE face. Its initial sub-horizontal curve has moved into a more oblique shape down to *c.* 30 m, forming a square angle *c.* 25 m below the surface and inclining to a sub-vertical shape more parallel to the SE surface in 1992. In 2015, the cold permafrost bodies have both totally disappeared

and the -1°C isotherm has retreated inside the rock mass forming a sub-rounded body. At that stage, the 0°C isotherm is parallel *c.* 5-10 m deep below the NW surface and *c.* 15-20 m below the SE surface.

At the AdM, the isotherms kept almost the same shape along the past 160 years, but warm permafrost has already penetrated at depth of the SE face in 1992, and reaches *c.* 15-20 m depth in 2015. The -5°C isotherm has narrowed into a *c.* 15 m wide in the shallow layers and parallel to the NW face in 1992. It has totally disappeared by 2015, whereas the -4°C has followed

the same pattern, subsisting between *c.* 3700 and 3820 m a.s.l. At this site, the *c.* 2°C warming between the end of the LIA and the recent period has affected the entire rock pillar.

Conversely, the central part of the GPA has remained unchanged along the past 160 years. But the -5°C isotherm, perpendicular to the SE face at *c.* 4130 m a.s.l. in LIA equilibrium conditions, has been affected by the climate change down to 30 m depth in 1992. It curves upwards to become more parallel to the SE face. In 2015, the -5°C isotherm has been

affected by the atmospheric warming down to *c.* 70 m and obviously retreats towards the NW face. Meanwhile the -4°C isotherm, initially located at *c.* 4050 m a.s.l. of the SE face, started to curve upwards following the same dynamic. In the NW face, isotherms have kept the same shape parallel to the rock surface. The shallowest 60 m with a temperature of -11 to -5°C in the LIA equilibrium conditions, has turned into a uniform body of -8°C into the entire NW face by 1992. Two decades later, this body has narrowed: its width decreased by 20 m at depth, its lower limit rose up to 4070 m a.s.l., and highest limit

lowered down to 4260 m a.s.l.

The air temperature rising experienced since the end of the LIA had variable effects depending on the site geometry and initial RST. The existing permafrost data in the Mont Blanc massif allow for an evaluation step of the model outputs at the current period.




## 4.2. Model evaluation

Modelled subsurface temperatures in rock walls are rarely benchmarked against measured borehole values due to the scarcity of subsurface temperature measurements, and because approved heat conduction models are assumed to be performant enough to reproduce accurate temperature in simple thermal systems such as rock wall permafrost. This assumption is valuable only if the upper boundary (the RST) is accurately simulated. Therefore, validation of modelled RST is often required before implementing heat conduction scheme, especially when it outcomes from complex energy balance simulations with many sources of uncertainties related to the high number of input data and parameters (Gruber et al., 2004; Noetzli et al., 2007). The quality of the initial RST was already mentioned in Section 3.2, based on the study from Magnin et al., (2015a). Here, we propose to evaluate simulated temperature at depth by mean of measured borehole temperatures and an electrical resistivity tomogram.

### 4.2.1. Evaluation with measured borehole temperature at the AdM

*Feflow* allows extracting model output at user defined observations points and time steps. We therefore requested extraction of simulated temperature for each observation point of the AdM model domain (see Sect. 3.2.2), and for each first day of each month between January 2010 and January 2015. Those modelled values are then compared to measured temperature in the AdM NW and SE boreholes to evaluate the model performances. Model output are first analysed at a daily time step before considering annual patterns. For better visibility, only four selected modelled temperatures of the year 2010 for each borehole, encompassing the four seasons, are displayed in Figure 6.

*Daily features*

The model reproduces very well the bedrock temperature below 6 m depth, especially in the NW borehole (Fig. 6a). Measured bedrock temperatures at shallowest depths are affected by snow cover, fractures and strong solar irradiation that are not taken into account in the modelling approach. This explains the greater discrepancy between modelled and measured ground temperature in winter (presence of snow) whereas summer temperatures better fit, especially in the NW face where the effect of solar radiation is weak.

The temperature profiles recorded by the NW borehole are significantly affected by an open fracture at 2.5 m depth which locally cools the bedrock due to air ventilation, especially during winter, whereas above the fracture, the insulating effect of a snow patch accumulating on a ledge at the borehole entrance warms the temperature profile down to the fracture depth (Magnin et al., 2015b). This is visible on the profile from the 1st January 2010 (Fig. 6a): the upper part shows small temperature gradient due to the snow insulation, while a stronger temperature gradient is visible below the fracture due to its shortcut effect between the air and the subsurface. Along the years 2010-2015, the maximum difference between measured and modelled daily temperature at 10 m depth in the NW borehole is 0.5°C and the mean difference of the 72 observation points (corresponding to 72 days between 2010 and 2015) is only 0.01°C.



In the SE borehole, the deepest measured temperatures seem less well reproduced by the model, but remain of reasonable accuracy with a maximum difference of 0.7°C is visible between measured and modelled value at 10 m depth along the observation period, and a mean difference of 0.1°C between the 60 measured points (two 3-4 months interruptions of the borehole records lower the number of available data) and simulated temperature at the same date. On the SE face, both the
almost continuous snow cover from fall to spring/early summer (Magnin et al., submitted) and the high solar irradiation affect the rock temperature, but are not considered in the modelling approach. The solar radiation effect is well visible on the measured profile from the 1$^{st}$ July (Fig. 6a).

*Annual features*

On an annual average, differences between the measured and modelled temperature values are smaller than on the daily basis
(Fig. 6b). At 10-m-depth, in the worst case (2010), the modelled value is 0.2°C higher than the measured one at the NW borehole, with a mean difference of 0.01°C along 2010-2015. At the SE borehole, the maximal difference between measured and modelled value is 0.04°C, but only two full years of observations (2010 and 2011) are available for model evaluation, the 2012-2015 time series being affected by significant gaps. Therefore, the mean difference between observed and measured value was not calculated.

The model reproduces satisfyingly the negative temperature gradient along the SE profiles, resulting of the heat sink effect of the opposite North face (Noetzli et al., 2007; Magnin et al., 2015b). On the NW face, the model shows a significantly lower temperature gradient than the measured one. Since 3D effects seem well reproduced on the SE face, the difference between the measured and modelled temperature gradient on the NW face may rather result from the cooling effect of the fracture than 3D effects.

Borehole temperature provides information at the point scale which is limited to evaluate 2D models forced with data having a metric resolution. However, they are much more suited than RST measurement points since the temperature at 10 m depth results from the heat transfer of a pluri-metric area at the surface. Further evaluation is permitted with electrical resistivity transect, which has been proven a relevant support to improve model evaluation (Noetzli et al., 2008).

**4.2.2. Evaluation with ERT at GM**

In October 2012, an ERT sounding was performed along a 160 m long profile of the GM NW face with an electrode spacing of 5 m (Magnin et al., 2015c). Field measurements were combined with laboratory testing on a Mont Blanc granite boulder in order to calibrate the resistivity-temperature relationship. Results allow for a semi-quantitative description of the permafrost state and suggested the presence of warm permafrost under the GM NW face. In Figure 7, the ERT transect is compared to the 2D temperature model of the GM for October 2012. The contour of the ERT transect is reported on the 2D
model by a red line. The active layer is represented by the positive temperature near the surface and the resistivity body < 80 kΩm corresponding to thawed granite. A warm permafrost body is delineated by the temperature in between -2 and 0°C and the resistivity between 80 and *c.* 200 kΩm. These features are visible on both sources of data and corroborate the model




performances in 2D representation of the permafrost. Further comments are limited by the lack of quantitative data: isotherms are not directly comparable to the iso-resistivities.

Given the remarkable capability of the transient simulations to reproduce current temperature conditions at the AdM borehole locations and in the GM NW face, the model can be judged performant enough to consider future long-term

scenarios.

### 4.3. Future scenarios

In Figure 8, time-dependent conditions for September 2099 in response to RCP4.5 and RCP8.5, assuming a +3 and +5°C increase in air temperature, respectively, are displayed. Future scenarios result in highly contrasted permafrost conditions, from an almost total disappearance (only relict body subsisting at the core) to preservation of entire permafrost conditions,

depending on both the RCP and the site.

At the GM, a relict body has subsisted in the internal part and below the topographical summit in both RCPs. Unlike the 20[th] century changes, with the isotherms retreating towards the NW face and in the interior of the summit, the permafrost body retreats downward in the core of the massif along the 21[st] century. Temperature gradients depend on the RCP, but are stronger than during the 20[th] century in both cases, with a difference of *c.* 4°C and *c.* 6°C between the shallow layers and the

deepest and core part for RCP4.5 and 8.5 respectively.

At the AdM, a *c.* 10-12 m wide body of cold permafrost (-3 to -2°C) still subsists under RCP4.5, located below the NW face and between 3710 and 3770 m a.s.l. All around, warm permafrost is largely present, and is found down to 7-10 m deep below the top and at *c.* 20-25 m depth of the central part of the SE face. Thus, permafrost has disappeared into the AdM SE face. In the most pessimistic scenario, permafrost has totally disappeared from the AdM summit, but warm permafrost will

still exist in the NW rock wall at *c.* 3750 m a.s.l. Similarly to the GM, the permafrost body retreats downwards.

At the GPA, the entire geometry has been affected by the projected air temperature increases. In the case of RCP4.5, cold permafrost is still largely present. Warm permafrost has penetrated into the SE face, reaching *c.* 40 m depth at *c.* 4100 m a.s.l. The -5°C isotherm has kept a similar curving shape than in 2015, but has crept towards the NW face. A *c.* 50 m wide colder body (< -6°C) still persists at depth of the NW face between 4050 and 4250 m a.s.l. RCP8.5 lead to a different

scenario: permafrost has disappeared in the shallowest 20-30 m of the SE face and warm permafrost exists at to *c.* 40-50 m depth of this face, with isotherms roughly parallel to the rock surface. The -5°C isotherm forms the coldest body which is *c.* 40 m wide, sub-rounded and located at *c.* 40 m depth of the NW face between *c.* 4060 and 4140 m a.s.l. In that case, the coldest areas are retreating in the core of the NW-half of the summit.

### 5 Discussion

In our study, we simulate long-term temperature evolution in three rock wall permafrost sites with different topographical settings. We use a relatively simple approach since transient simulations are only governed by air temperature changes



applied to the RST and transferred at depth with the only consideration of heat conduction and latent heat exchange processes. Such simplifications in the calculation of RST changes ignore complex heat exchanges driving the energy balance at the rock wall surface and driving the heat transfers at depth in the natural environments. This results in a certain degree of uncertainty in the model output. However, such a simplified approach is permitted in steep rock walls due to the relatively

simplicity of the thermal system, and bears the main advantage to facilitate the identification of uncertainty sources, conversely to thermal models with complex energy transfers and multiple feedbacks. The limits in our modelling approach are first examined prior to discuss the implication of our results for determining future permafrost changes in the steep rock walls of the Mont Blanc massif.

## 5.1. Model limits

Model uncertainties arise from misrepresentation of the processes in the thermal system, unknown physical properties and errors in input data (Gupta et al., 2005; Gubler et al., 2013). Regarding the modelling approach adopted in this study, uncertainties mainly arise from five different sources.

### 5.1.1. Initialisation and forcing data

Assumptions and simplifications were necessary to generate the initial 2D temperature field at the end of the LIA and run

transient simulations. Starting with LIA equilibrium conditions ignores possible transient effects from Würm and Holocene climate variability. But a thermal perturbation at depth due to millennial time-scale changes is unlikely given the geometry of the investigated sites (max. width of 350 m), and results from previous studies (Kohl, 1999; Noetzli and Gruber, 2009). Simulated 2D temperature fields seem accurately reproduced by the modelling procedure. Their comparison to real-world data with the AdM borehole temperatures and the GM ERT transect underscores two strong points. Firstly, the closeness

between measurements and model (Sect. 4.2.2) in the uppermost layer of the rock walls (*i.e.* the 10 and 25 shallowest meters at the AdM and the GM, respectively) suggests that the strategy to force transient simulations is relevant enough to simulate the permafrost changes since the LIA up to the current period. In the meantime, it emphasizes the quality of the forcing data that have the advantage to be partly built upon direct measurements of air temperature in the Mont Blanc area. This ensures a better representativeness of the local climate variation compared to commonly used km-scale grids data set to simulate local

distribution of alpine permafrost.

The resolution of the input data is one of the most challenging issues to force permafrost models in highly heterogeneous land surfaces such as mountain permafrost (see next Section; Gruber, 2012; Fiddes et al., 2015). Downscaling methods are under development for mountain terrains (Fiddes and Gruber, 2012; 2014; Fiddes et al., 2015), but currently available km-scale data set remain better suited to drive numerical model on larger areas with a coarser spatial resolution (Gruber, 2012;

Jafarov et al., 2012; Westermann et al., 2013; Gisnas et al., 2013). These approaches generally consider the most prominent controlling factors of the regional permafrost distribution, such as air temperature or precipitations, but are not representative of smaller-scale heterogeneities induced by additional factors such as the solar radiation.





More sophisticated approaches capable of simulating complex energy exchanges at the bedrock-atmosphere interface have been proposed to model surface temperature of steep mountain slopes using specific algorithms to compute solar radiations (Stocker-Mittaz et al., 2002; Gruber et al., 2004; Salzmann et al., 2007). They provide information on the aspect-specific changes such as higher sensitivity of the North-facing slopes to future climate change, or a higher variability in the rate of

change of the South-exposed faces. But it has to be kept in mind that such complex approaches induce a high degree of uncertainty due to the numerous parameters and feedbacks owning their respective sources of error. In consequence, a certain proportion of the modelled RST variability between different model outputs is in the range of the model noise (Salzmann et al., 2007).

Finally, the energy balance at the rock surface is further influenced by the intermittent and heterogeneous presence of snow.

Current research developments focus on both the quantification of snow deposits onto steep slopes (Wirz et al., 2011; Sommer et al., 2015), and the modelling of its impact on the rock surface temperature (Pogliotti, 2011; Haberkorn et al., 2015a; 2015b) or on the active layer thickness (Magnin et al., submitted). But relevant data and parameterization technics are still lacking to drive long-term transient models of rock wall permafrost accounting for snow effects.

### 5.1.2. Future scenarios

In this study, possible climate evolutions over the future decades are obtained from one single climate model (namely, IPSL-CM5A-MR) and two scenarios for greenhouse gas emissions, out of four possible RCPs, and 20 to 40 models (depending on the model versions and the type of the climate simulations) that participated to the 5[th] Assessment Report of the IPCC (2013). The choice of the climate model was motivated by its realistic steady-state in present-day conditions when compared to observational time series, its median response to greenhouse effect evolutions, as well as its rather high spatial resolution,

ensuring more realistic topographic effects. The retained radiative forcings for the 21[st] century consist of the so-called RCP4.5 and RCP8.5, that is, a moderately optimistic and a pessimistic scenario (see Section 3.3.2, Moss et al. 2008). These RCPs can be considered, today, as reasonable estimations of the lowest and highest air temperature changes likely to happen, and thus, plausible contrasted boundaries within which the permafrost could evolve. According to the Paris agreement on climate change adopted on December 2015 during the COP21, climate change should be limited "well below 2°C" above

pre-industrial levels with more ambitious targets updated every five years. To date however, the reductions in greenhouse gas emissions planned by the participating countries lead to an estimated global warming of roughly 2.5-3°C, which lead us to discard the more optimistic RCP2.6 scenario, describing a rapid decrease in emissions as soon as the early 2020s, but makes RCP4.5 the most probable pathway for the 21[st] century.

### 5.1.3. Model dimensions and resolution

Rock wall permafrost is highly sensitive to climate change since its signal penetrates throughout the different sides of a summit (Noetzli et al., 2007). In this study, we approach rock wall temperature changes in 2D only. Therefore, the simulated changes in permafrost distribution are possibly slightly under-estimated since the signals only penetrate through two sides



(NW and SE). However sensitivity analyses have shown that bi-dimensional simulations show similar long-term transient temperature pathways than in 3D situations (Noetzli and Gruber, 2009). Based on these previous findings, it can be assumed that the 2D temperature fields are acceptable to draw reasonable patterns of permafrost distribution and changes.

Also, the model resolution (4 m), defined by the initial RST resolution (Sect. 3.2.1), later refined for spatial discretisation of the model domain (Sect. 3.2.2), is sufficient to represent the main topographical control on the rock wall temperature distribution, and to drive long-term changes. Permafrost modelling often suffers of coarse topographical resolution which does not represent the natural variability of environmental parameters (Etzelmuller, 2013). Metric resolution is essential for realistic simulation of rock wall permafrost at the summit scale (Magnin et al., 2015a). The satisfying quality of the 2D model outputs for the current period (Sect. 4.2.1) confirms that the model resolution is enough to address long-term permafrost changes in the selected sites. The comparison of measured and modelled temperature profiles indicates that the bedrock temperature at depth, which results from the heat transfer in a pluri-metric catchment, is mainly a response to air temperature changes and conductive heat transfer. Simulations at shorter spatial and temporal scales, especially in the uppermost layers, would certainly require higher spatial resolution of the model domain and consistent input data.

### 5.1.4. Thermal parameters

Subsurface thermal parameters have been defined upon published values for hard rock. The thermal conductivity and heat capacity are assumed homogeneous in the model domain whereas they vary with frozen/thawed conditions in the natural environment due to the changing properties of the interstitial ice/water. The variable state of the interstitial ice/water results in variable thermal conductivity of the porous and saturated rock media along seasons (Wegmann et al., 1998) and longer term permafrost changes. This is accounted for in our modelling approach throughout Equation (2).

We selected conservative values for granitic rock but tested the model sensitivity to different thermal conductivity values (2.7 and 3 W.m$^{-1}$.K$^{-1}$). Results confirmed findings from previous studies: a lower conductivity increase the geothermal heat flux control (Maréchal et al., 2002; Noetzli et al., 2007) but did not lead to substantial changes (Kukkonen and Safanda, 2001). Furthermore, the thermal conductivity is naturally anisotropic (Goy et al., 1996), whereas it is considered isotropic in our modelling approach. Increasing the conductivity in horizontal directions increases the topo-climatic control, whereas increasing the conductivity in vertical direction gives more importance to the geothermal heat flux (Noetzli and Gruber, 2009).

The influence of the geothermal heat flux appears of relatively high importance for running steady-state conditions. Equilibrium conditions without the influence of the upwards and deep-seated flow only depends on the climate control, which can lead to highly different conditions than when balancing the respective influences of the geothermal heat flow and the climate. In Figure 9, an example of the geothermal heat flux effect is given for the GM, which is the most sensitive site to this parameter given its relatively low relief and wide geometry. Without geothermal heat flow, the equilibrium condition for the LIA shows almost entirely cold permafrost conditions with more vertical isotherms. However, when running transient simulation, the influence of the geothermal heat flux is less significant on the temperature range. It only affects the shape of



the isotherms at the summit basis, leading to permafrost retreat in the core of the summit instead of only lowering down. Simulations without geothermal heat flux were also run with a lower thermal conductivity (2.7 W.m$^{-1}$.K$^{-1}$) and a monthly time step, which had no impact on the results of this study (Fig. 9): only the uppermost layers show different temperature range due to different time step in the forcing data (daily *versus* monthly), but this is far beyond the scope of this study.

Idealized test cases suggested that mountain summits are decoupled from the deep-seated geothermal flow influence (Noetzli et al., 2007), but such settings are not representative of most of the alpine study cases, more or less wide and more or less elevated.

### 5.1.5. Heat transfer processes

Energy transfers inside the rock mass are mainly driven by heat conduction processes, whereas fluid flows can be, in a first
approximation, neglected to simulate long-term changes (Kukkonen and Safanda, 2001). Nevertheless, advective heat transport by water circulation along fractures may locally warm the bedrock at depth (Hasler et al., 2011a). Conversely, air circulation in open clefts would rather cool the bedrock (Hasler et al., 2011b; Magnin et al., 2015b). These non-conductive heat transfers are not accounted for in the here presented modelling approach. The evaluation step against borehole temperatures clearly shows their effect in the shallow layer (the *c.* 6 upper meters below the surface, Sect. 4.2.1) while
temperature at deeper layers is accurately represented with the only consideration of 2D heat conduction.

The melting of ice bodies in the fractures may also be expected in the natural environment, such as suggested by recent observations in rock fall scars (Ravanel et al., 2010b). Ice melting may significantly dampen and lower the rate of temperature changes at depth by the consumption of latent heat (Wegmann et al., 1998; Kukkonen and Safanda, 2001). Some ice-filled fractures can turn into thawing corridor during the thawing season (Krautblatter and Hauck, 2007), which can
favours the melting of ice-filled fractures, and degrades rock wall permafrost in unexpected areas and depths (Hasler et al., 2011a). Such processes were approximated with a relatively high porosity value in the model domain (5 %), which fails in representing the anisotropic and heterogeneous character of such processes. Further developments are highly encouraged to gain data on rock wall structure, especially with the use of geophysical soundings, which will not only support improvement of thermal modelling in alpine topography, but will also contribute in bridging the gap between thermal, hydraulic and
mechanical models for future risk assessment (Krautblatter et al., 2012).

### 5.2. Past and future permafrost degradation

Taking into consideration the model limits, the mid-term changes along the 21$^{th}$ century can not be reliably interpreted, whereas the interpretation of short-term changes is beyond the scope of this study. However, long-term permafrost changes can be considered despite the shape of the isotherms remains partly uncertain due to limitations in the thermal parameters. In
this section, we summarize the permafrost changes since the end of the LIA to the current period. Then, we examine possible changes by the end of the 21$^{st}$ century. The patterns of simulated 2D permafrost degradation from the termination of the LIA



to the end of the 21$^{st}$ century show dependency to the topographical settings (summit width), to the bedrock temperature (latent heat effects), and to the intensity of the climate signal.

### 5.2.1. Permafrost degradation since the LIA termination

The rate of change between equilibrium conditions in 1850 and early 1990s was in the same range as the one experienced during the past three decades. Indeed, for the first time period, the thermal perturbation has been detected down to *c*. 30 m below the surface (depth at which the equilibrium isotherms have been affected), whereas it reaches at least twice deeper layers in 2015. Narrow peaks like the AdM have been entirely affected by the air temperature increase since the end of the LIA due to the short distance between both sides from which the signal penetrates, and the resulting intense lateral heat fluxes, especially at the top (Noetzli et al., 2007). The core of wider geometries, where N and S-facing surfaces are both distant by more than *c*. 150 m, such as the GPA and the GM, has remained intact.

Temperature changes during 1850-2015 were not as high at the GM as in the two other sites, and were not as high as the air temperature change, because the entire summit was in the temperature range within which latent heat exchanges occur. This pattern is aligned with the global trend showing the delaying effect of latent heat uptake in close to 0°C boreholes (Romanovsky et al., 2010). But the rate of temperature change strongly increases during the 21$^{st}$ century when the GM temperature becomes almost entirely positive.

In between the 1990s and 2010s, permafrost has disappeared and warm permafrost has largely penetrated in the shallow layers of the GM and AdM SE faces, respectively. Quantitative interpretations about the permafrost lower boundary and its changes are limited by the discontinuous character of mountain permafrost mainly governed by local conditions (Etzelmuller, 2013), topographical and transient controls, which respective influence is difficult to distinguish (Noetzli and Gruber, 2009). Nevertheless, results of our 2D simulations clearly suggest that climate change in between the LIA termination and the 2010s, especially since the 1990s, have lead to permafrost disappearance below the S-exposed faces at least up to 3300 m a.s.l. (top of the GM), but not above 3700 m a.s.l. (foot of the AdM SE face). Thus, lower boundaries of snow-free and hard rock wall permafrost lie within this elevation interval, but Magnin et al. (2015a) have suggested that isolated permafrost bodies could exist down to 2800 m in favourable S-exposed slopes where conduction is not the prominent heat transfer process (due to high fracturing for example). Not taking into account snow and fracturing parameters could lead to a 3°C under-estimation of bedrock temperature is S-exposed rock walls (Hasler et al., 2011b).

Warm permafrost has thickened below the S-exposed face at up to 3850 m a.s.l. during the past two decades meanwhile it was extending below the N-facing slopes up to 3300 m. Whether warm permafrost reaches *c*. 20 m depth below the S-exposed AdM face in 2015, this depth can not be extrapolated to other sites due to the site-specific effect of the opposite N face.



### 5.2.2. Permafrost degradation during the 21st century

By the end of the 21st century, even the core of the relatively wide summits such as the GPA will be strongly affected by the projected increase in air temperature along the 21st century. Warm permafrost will extent at depth of the S-exposed faces at least up to *c.* 4300 m a.s.l, and at depth of the N-facing faces lower than 3850 m, according to the RCP4.5. Permafrost will

certainly disappear in all aspects below 3300 m, and up to 3850 m a.s.l. at least (top of the AdM) in the S-exposed faces. Following the most pessimistic scenario, permafrost will disappear in the subsurface of the S-exposed rock walls at least up to 4300 m a.s.l. (top of the GPA), while cold permafrost will still exist at same elevation in N-facing slopes if the S face influence does not affect it. At lower elevation such as at the AdM, warm permafrost will still occur below the N-exposed faces, but could disappear in the narrowest sections due to the S-facing slope influence. When both mountain sides do not

allow for permafrost conditions any more, such as the GM, the permafrost body will retreat downwards, in the core of the summit.

These degradation patterns locally depend on the topographical control, especially the summit width which either reinforces the intensity of the climatic signal in narrow geometries by mixing S-facing and N-facing slope influences, or maintains independency between the shallow layers of the S and N-exposed slopes when it is wide enough (pluri-hm). Site-specific

patterns make the concept of "lower limit" not adapted to describe rock wall permafrost distribution and changes.

### 6 Conclusions and perspectives

Former studies have described the 3D processes and patterns of permafrost degradation in idealized high mountain geometries of the European Alps following former IPCC reports (Noetzli et al., 2007). In this study, we investigated permafrost degradation from the LIA steady-state until the end of the 21st century in three summits representative of the

topographical and permafrost settings of the Mont Blanc massif rock walls, and with the most recent climate models. Simulations are performed in vertical cross-sections with local air temperature forcing data and two possible air temperature scenarios, accounting for moderately optimistic and pessimistic 21st century pathways. They provide insight in the past and future changes experienced by rock wall permafrost in the Mont Blanc area, relevant for geomorphological applications. The main outcomes are the following:

1. Thermal conditions for the current period (2010-2015) are remarkably well represented when comparing simulated temperature field to independent data set. Our modelling approach is therefore well appropriate to run long-term transient simulations in rock walls.

2. Thermal perturbation induced by climate change since the end of the LIA is visible down to *c.* 60-70 m below the surface, that is, the narrow AdM peak has been entirely affected.



3. In between the 1990s and the 2010s, (i) permafrost has disappeared in the shallowest *c.* 20 m of the S-exposed faces and warm permafrost has extended at depth of the N-exposed faces at least up to 3300 m a.s.l., (ii) warm permafrost has extended at least up to 3850 m and penetrated down to 20 m depth of the S-exposed faces at *c.* 3700-3800 m a.s.l.

4. At the end of the 21$^{st}$ century, only relict permafrost body will persist in the core of the wide summits below 3300 m a.s.l.

5. Considering a moderately optimistic scenario (RCP4.5), permafrost will disappear along the 21st century in any topographical settings lower than 3300 m a.s.l., and at depth of the S-exposed faces up to 3850 m. Warm permafrost will extent below the N-exposed faces at similar elevation and at least up to 4300 m a.s.l. below the S-exposed faces. Cold permafrost will still exist below the N-exposed faces above 4000 m.

6. Considering the most pessimistic scenario, permafrost will disappear of the S-exposed faces at least up to 4300 m a.s.l., and lead to permafrost disappearance in the narrow summits below 3850 m. But without the influence of a close S-exposed face, warm permafrost could persist at least down to 3700 m a.s.l. in the N-exposed faces. Cold permafrost will still exist, in spite of significant warming, at depth of the N-facing slopes higher than 4000 m a.s.l.

7. The simulations show that the patterns of permafrost changes are mainly governed by the local topographical control, which underscores the specificity of rock wall permafrost and restricts the extrapolation of the results.

8. Transient simulations provide useful information for analysis of the thermal conditions at rock fall locations, but analysis of specific event would require shorter time-scale simulations with consistent input data and thermal parameters.

These results provide useful information to analyse the link between rock wall thermal characteristics and rock fall triggering. Future development will use these results for a detailed analysis of the link between permafrost changes and the inventoried rock fall in the Mont Blanc massif. Future simulations will involve 3D modelling, shorter-time scale simulations in anisotropic and fractured media, with non-saturated conditions and fluid transfers, as well as simulations with downscaled climate data. Those developments will constitute essential steps for bridging the gap between thermal and mechanical models.

**Data availability**

Data are available upon request to the first author.

**Author contribution**

Florence Magnin conducted the study. She designed the modelling approach, prepared the input data, performed the model evaluation, interpreted the results, and wrote the manuscript with inputs from the co-authors. Jean-Yves Josnin implemented the modelling approach in Feflow, performed the model runs and contributing in writing of the methods. Ludovic Ravanel is the responsible of the WP Permafrost in the *VIP Mont Blanc project*, and contributed in designing the study and writing of the manuscript. Benjamin Pohl and Julien Pergaud extracted the climate forcing variables, contributed in the choice of the RCP forcings and in writing of the methods and the discussion. Philip Deline contributed in writing of the manuscript.





## Acknowledgments

This work was founded by the French *Agence Nationale de la Recherche* in the scope of the *VIP Mont Blanc* project (ANR-14-CE03-0006) and by the EU Interreg V-A France-Italy ALCOTRA 2014-2020 n°342 *PrévRisk Haute Montagne* project. We acknowledge Peter Schätzl and Carlos Rivera from Feflow's staff for their help in our beginning with the Pi-Freeze plug-in.

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



**Figure 1: Location of the Mont Blanc massif, its glaciers and mean annual rock surface temperature (MARST). Areas with MARST < 0°C can be considered as permanently frozen (Magnin et al., 2015a). Background topography: ASTER GDEM.**





**Figure 2: Topographical profile locations on the three sites.**




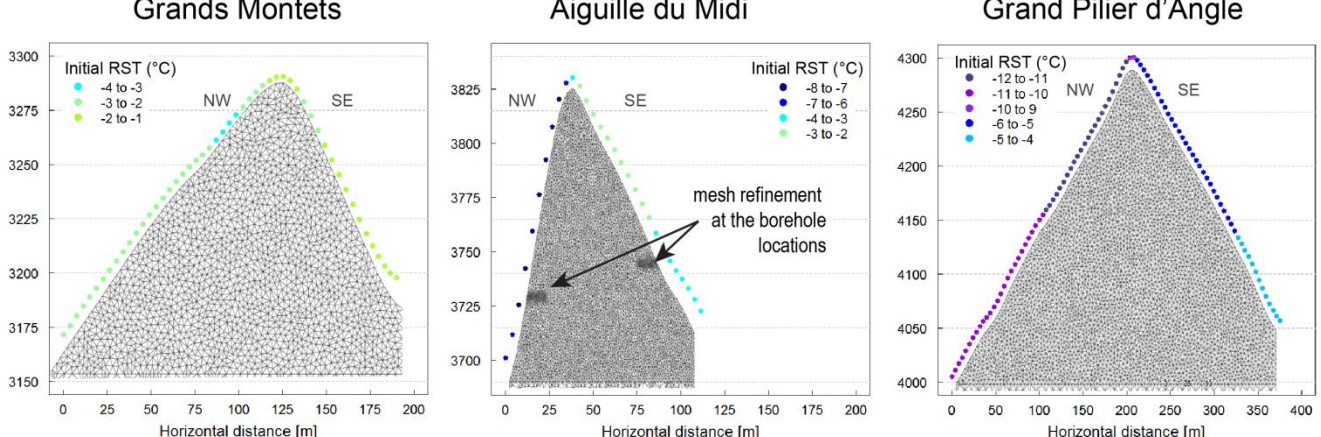

**Figure 3: Boundary conditions: initial RST plotted over the models meshes for the three sites. The spatial resolution of the initial RST (coloured dots) has been refined at the mesh scale by linear interpolation. Below the topographies, a box of 5000 m elevation and a constant geothermal heat flux of 85 mW$^{-2}$ were set up.**

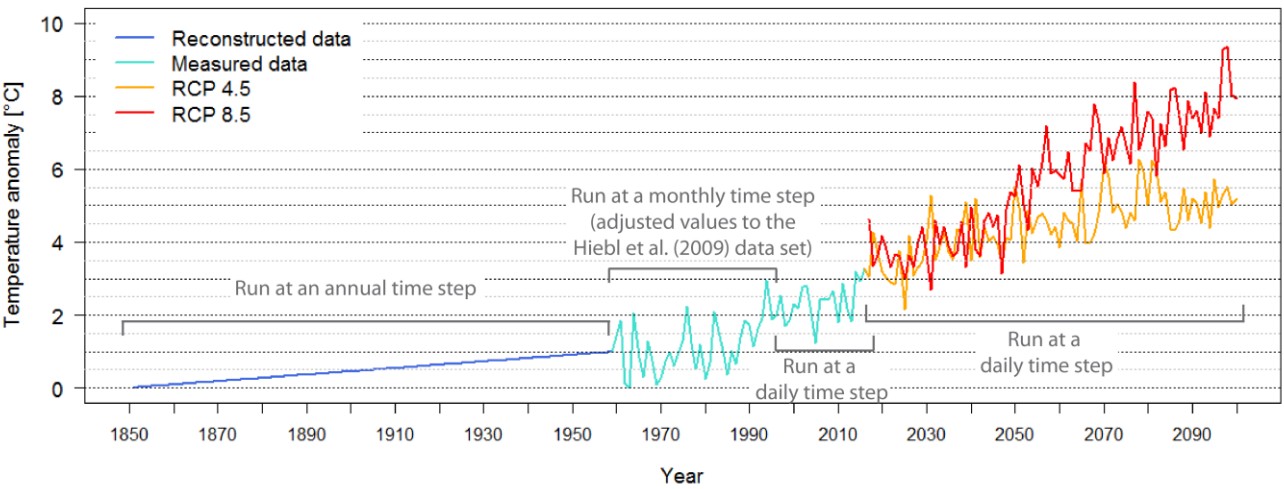

**Figure 4. Forcing air temperature data displayed at an annual time step. For running the transient models, the time step was refined for some periods as described on the figure.**





**Figure 5. Initial steady-state (a) conditions and time-dependent conditions in September 1992 (b) and September 2015 (c) for the three studied sites (note that figure scales are different).**



**Figure 6. Evaluation of modelled bedrock temperature against measured bedrock temperature in the AdM NW and SE boreholes at daily time-step (a) and annual time-step (b).**



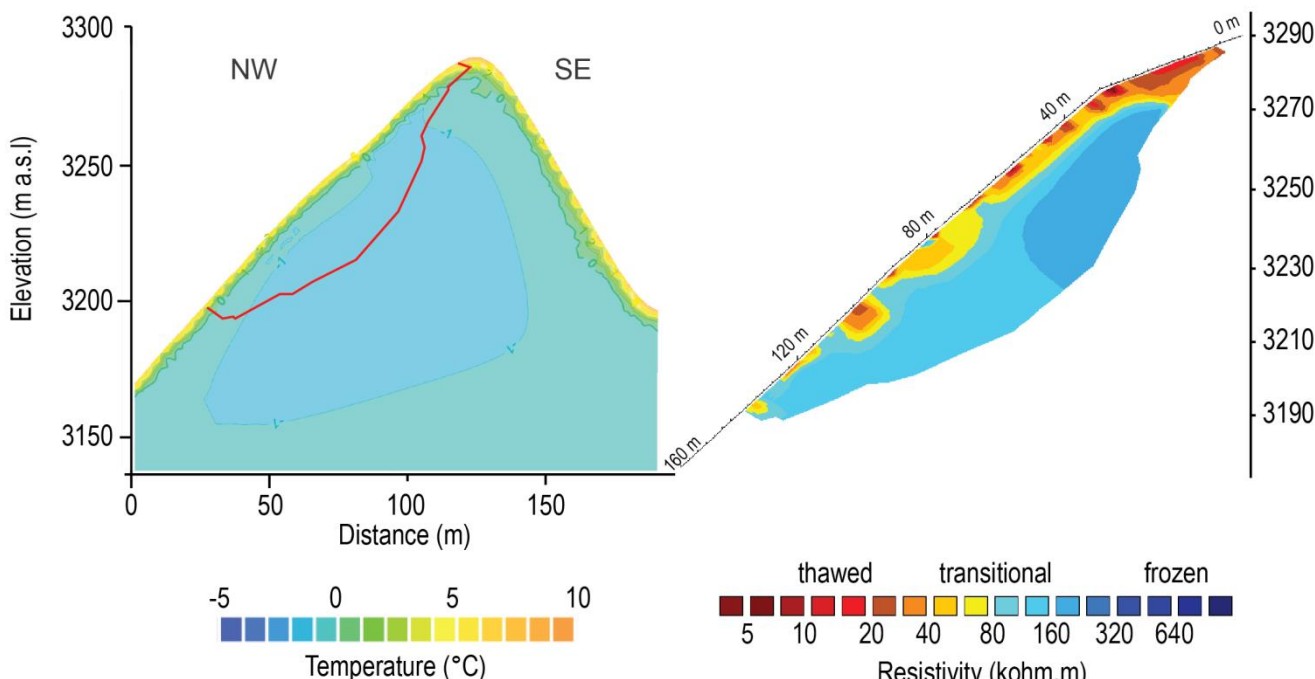

**Figure 7. Model evaluation (left) against ERT transect (right) for October 2012 at the GM. The red line on the left figure**
5  **delineates the ERT contour.**



**Figure 8. Time-dependent conditions in September 2099 after RCP4.5 and 8.5 forcings for the three investigated sites.**





Figure 9: Bi-dimensional thermal fields at the GM site for steady-state conditions (1850) and transient conditions (2010). Models on the left are run at daily time step with a geothermal heat flux (GHF) of 85 mW.m⁻² and a thermal conductivity of 3 W.m⁻¹.K⁻¹, whereas those on the right are run at monthly time step without geothermal heat flux and with a thermal conductivity of 2.7 W.m⁻¹.K⁻¹.