# Peer review of "Modelling rock wall permafrost degradation in the Mont Blanc massif from the LIA to the end of the 21st century"

_The Cryosphere, 2016_

## Referee Comment (RC1) · Anonymous Referee #1 · 13 Oct 2016

This paper investigates the rock wall permafrost evolution from 1850 (Little Ice Age) to 2100 in Mont Blanc massif. Applying a simple 2D numerical model based on heat conduction equation and latent heat transfer, this study shows the possible ongoing degradation of permafrost at representative selected locations according to two climatic scenarii. The results outline local topographical control on the patterns of permafrost changes, and evidenced a general trend to increased permafrost degradation at high altitude.

I found this paper very clear and well written. The purpose of the work is well articulated and methodology adapted. The collected data (temperature and ERT) in such harsh environmental conditions are impressive and are well used to evaluate model

performance. Although simple, the applied numerical model appears to be quite efficient to simulate the long term temperature evolution within the three selected rock walls. The discussion section honestly presents the model limits.The scientific results and conclusions are presented in a clear, concise, and well-structured way. Therefore I will have only few comments.

However, I am not convinced this study constitutes a substantial progress in current scientific understanding: Although the results try to quantify and map the recent and future changes of rock wall temperatures (according to climatic scenarii based only on temperature), the conclusions are of no surprise considering initial assumptions and model limitations. I was unable to find really new insights on permafrost study. Although authors claim this study provides insights for retrospective stability analyses of rock walls (which appears to be the main motivation of the paper), they never give a way to tackle this very important problem (for example characterizing temperature gradient within rock wall, linking fracture dynamics to temperature changes, quantifying effect of saturation on temperature changes,...). Moreover, as rock wall stability might be essentially driven by rapid external/internal changes (solid/liquid precipitation, air temperature, permafrost evolution, fracture propagation, previous rockfalls,...), I doubt this long term approach would help for stability assessment (as this approach cannot "debate on short-time scale" and on 3D effects due to limitation in the modelling approach).

**General comments:**

This study only accounts for possible temperature changes, not for the solid/liquid precipitation changes. Even if the model do not consider precipitations (snow or rain), precipitations might really influence the global permafrost pattern - presence of snow patches, liquid precipitation percolating into surface fractures,... Even if total annual precipitations might not evolve drastically in the future, a change in seasonality might occur, leading to a different ratio between solid and liquid precipitations. Even if this

effect is marginal, this might be mentioned in introduction section and in section 5.1.2

The model section could be slightly reorganised: After exposing modelling strategy, I was expecting the description of the core of the model, i.e. the heat transfers. I suggest to first describe how heat transfers are modelled before describing boundary conditions and transient simulations (as both refer to heat transfer computations).

The description of the numerical approach is not clear:
1. Equation 1 refers to conservation of energy WITHOUT phase transition (heat capacity). The presence of a phase transition might introduce a discontinuity. How exactly is the phase transition treated? Please provide additional explanations.
2. How/why exactly is used the freezing function (eq. 3)? The definition is clear but the use in the calculation is not clearly define. Is this relation directly given by the model, or empirical? Is it meant to define the amount of ice formed during the phase transition? Where is latent heat in your model?

**Specific comments:**

In abstract:
page 1 line 12: define LIA in abstract.
p. 1, l. 16: describe briefly the model used (2D Finite Element Model accounting for heat conduction and latent heat transfer).

p. 3, l. 12: again describe a bit the model.
p. 4, l. 9: mechanics? please develop (is it not fracture kinematics?).
p. 4, l. 12 and in the following: abbreviation *c.*: I though such abbreviation was used only for dates.
p. 5, l. 19 and l. 29: I would suggest to redefine here RST and MARST for readers convenience.
p. 9, l. 25-29: Not clear: adaptable thermal properties of ice? configurable latent

heat??? This should be constant, please elaborate.

p. 10, l. 6: Is this 5% value constant in the whole domain? Why not setting high porosity near the surface to somehow represent the presence of fractures and a smaller one in the interior of the wall?

Figure 2: < -5 in colormap caption. Add (a) detailed MARST and (b) topographical situation.

Fig. 6: I would suggest to take another kind of marker (circle and bullet are too similar, and sometimes cannot be distinguished).

---

## Referee Comment (RC2) · Anonymous Referee #2 · 28 Oct 2016

The article by Magnin et al. entitled "Modelling rock wall permafrost degradation in the Mont Blanc massif from the LIA to the end of the 21st century" addresses an important subject relevant to various stake holders in high alpine environments. The authors have collected valuable data by undertaking considerable field work in a harsh environment. They also present 2D model results of long term simulations of the thermal state of three mountain top permafrost sites as well as one ERT transect of the north-west facing wall of the Grands Montets.

However, from my point of view the work does not make a significant contribution neither to permafrost modelling nor to the understanding of high mountain permafrost. The language is confusing throughout large parts of the manuscript. Further, the dif-

ferentiation the authors seem to make between rock wall permafrost and high mountain permafrost in general - which is especially evident from the choice of the cited literature - is not convincing to me. Are there different hydro-thermal processes acting in rock walls than at less exposed sites or are they just active in different proportions? It is just stated that rock wall permafrost "is a relatively simple system" and it is further assumed to be homogeneous and saturated. The simplified assumption, that radiation and air temperature are sufficient for the simulation of such a system, especially if one of the relevant research questions in this context would be to gain insights on the triggering of future rock fall events (which are almost certainly not just a thermal phenomenon and would not occur in a homogeneous matter) - in my eyes - is to be considered as inadequate. It is also stated, that only air temperature was used to drive the model, which is in contradiction to this already simplified assumption. The question also arises where the water is coming from, or going to, when freeze-thaw processes occur in the model?

I would encourage the authors to rewrite and restructure large parts of the manuscript and to address these questions as well as to refer to additional relevant literature on the subject. The uncertainties arising from the chosen simplifications should be discussed in this light and the conclusions drawn should also reflect that.

---

## Author Comment (AC1) · 16 Nov 2016

Dear Handling Editor, dear reviewers,

All the authors deeply thank you for your review, questions, comments and suggestions.

All these remarks have been considered to revise our paper. We hope that it now satisfies the standards for a publication in The Cryosphere special issue on the Evolution of mountain permafrost. In the revised version the major changes are the following:

1.Presented and discussed the interests and limitations in using precipitation scenarios

2. Reorganised the sections presenting the modelling approach

3. Added a sub-section in the Discussion to consider how the presented results and methods would be used for rock wall stability analysis.

Hereafter you will find the detailed answers to your comments. A revised version with change tracks is provided in order to follow our work. We replied to every comment and question.

Best regards.

Dr. Florence Magnin, on behalf of all the authors.   Answers to reviewer's comments

Reviewer 1

1.However, I am not convinced this study constitutes a substantial progress in current scientific understanding: Although the results try to quantify and map the recent and future changes of rock wall temperatures (according to climatic scenarii based only on temperature), the conclusions are of no surprise considering initial assumptions and model limitations. I was unable to find really new insights on permafrost study.

Authors answer: We agree that this paper does not provide new methodological development to investigate long term evolution of rock wall permafrost. The aim of our paper is (i) to enhance recent developments in permafrost modelling by using the existing tools, procedures, and data, (ii) to show up the capabilities of simple modelling approach in reproducing rock wall thermal fields, which, in existing studies, was not substantially demonstrated against field data, and (iii) to use the climate projections obtained from the latest climate models driven by the most recent scenarios to (iv) propose quantitative and site-specific permafrost pathways for the future. Therefore the aim was not to provide new insight in permafrost studies but new insight in high mountain permafrost future changes, and we hoped that this topic would fit in "The evolution of permafrost in mountain regions" special issue. Such impact models are essential to anticipate the future dynamics of permafrost and, as mentioned by Reviewer 2, are extremely relevant for stakeholders.

2. Although authors claim this study provides insights for retrospective stability analyses of rock walls (which appears to be the main motivation of the paper), they never give a way to tackle this very important problem (for example characterizing temperature gradient within rock wall, linking fracture dynamics to temperature changes, quantifying effect of saturation on temperature changes,...). Moreover, as rock wall stability might be essentially driven by rapid external/internal changes (solid/liquid precipitation, air temperature, permafrost evolution, fracture propagation, previous rockfalls,...), I doubt this long term approach would help for stability assessment (as this approach cannot "debate on short-time scale" and on 3D effects due to limitation in the modelling approach).

Authors answer: We agree that the paper does not discuss in details the possibility of such thermal models for retrospective analysis of bedrock detachment. Therefore in the revised version we discuss more deeply this topic to strengthen the interest of our paper (new Section 5.3). Because the paper is already substantial and that the main topic addresses permafrost future changes, we don't develop additional modelling applications for stability assessment, which is the content of a study that we are currently developing (working on saturation changes and hydrostatic pressures). We disagree on the fact that such long term models can't be used for stability assessment. This study shows for the first time that the model simulates realistic temperature fields which is a preliminary step in stability assessment. This is discussed in more details in the revised version. Also, we removed the following confusing sentence in the abstract "Shorter time-scale changes are not debatable..." which is more accurately expressed in Section 5.2. "...interpretation of short-term changes is beyond the scope of this study". Given the evidences that even at daily time step temperature thermal fields are well reproduced, short time scale changes are debatable but beyond the scope of this study. However, they are worth to discuss for retrospective analysis of rock wall destabilisation.

3. This study only accounts for possible temperature changes, not for the solid/liquid

precipitation changes. Even if the model do not consider precipitations (snow or rain), precipitations might really influence the global permafrost pattern - presence of snow patches, liquid precipitation percolating into surface fractures,... Even if total annual precipitations might not evolve drastically in the future, a change in seasonality might occur, leading to a different ratio between solid and liquid precipitations. Even if this effect is marginal, this might be mentioned in introduction section and in section 5.1.2.

Authors answer: Indeed, precipitation control is assumed to be marginal on the long term change of rock wall permafrost. But we agree that this might deserve more consideration in our introduction and discussion. Therefore, we reworked the introduction and section 5.1.2 accordingly in the revised paper. In the introduction we only mentioned the possible role of water percolation along fractures for permafrost degradation and rock fall triggering and why we only consider air temperature in the forcing data. In the discussion we developed a little bit more the role of precipitations. Anyways, integrating precipitations would not make sense in our study because this is depending on local processes that global circulation model do not represent. Their resolution is too coarse compared to the scale of our study sites. Also, at high elevation, most precipitations are solid and proper simulation of their accumulation patterns on steep slopes is still challenging since it depends on steepness, roughness, wind- and sun-exposures. Integrating precipitation in such models is a nice perspective but this is not really feasible at the moment.

4.The model section could be slightly reorganised: After exposing modelling strategy, I was expecting the description of the core of the model, i.e. the heat transfers. I suggest to first describe how heat transfers are modelled before describing boundary conditions and transient simulations (as both refer to heat transfer computations).

Authors answer: Yes, this is a possible way to organise the section, therefore we reworked the revised version as suggested here above.

5.The description of the numerical approach is not clear: Equation 1 refers to conservation of energy WITHOUT phase transition (heat capacity). The presence of a phase transition might introduce a discontinuity. How exactly is the phase transition treated? Please provide additional explanations.

Authors answer: We took this comment into consideration and reworked the text accordingly. The discontinuity induced by the phase changes is detailed in Equations 4 and 5 of the revised version that explain how thermal conductivity and heat capacity are modified compared to Equation 1.

6. How/why exactly is used the freezing function (eq. 3)? The definition is clear but the use in the calculation is not clearly define. Is this relation directly given by the model, or empirical? Is it meant to define the amount of ice formed during the phase transition? Where is latent heat in your model?

Authors answer: Yes the Equation defines the amount of ice/water formed during the phase change. The latent heat is taken into account with Equation 5 of the revised version. We reworked all the section in order to better address these questions

Specific comments: In abstract: 1. page 1 line 12: define LIA in abstract. Authors answer: Done

2. p. 1, l. 16: describe briefly the model used (2D Finite Element Model accounting for heat conduction and latent heat transfer). Authors answer: Done

3. p. 3, l. 12: again describe a bit the model. Authors answer: Done

• p. 4, l. 9: mechanics? please develop (is it not fracture kinematics?). Authors answer: Done

4. p. 4, l. 12 and in the following: abbreviation c.: I though such abbreviation was used only for dates. Authors answer: True, this abbreviation is more commonly used for date. We replaced these abbreviations when it was not used for dates in the revised version.

5. p. 5, l. 19 and l. 29: I would suggest to redefine here RST and MARST for readers convenience. Authors answer: Done

6. p. 9, l. 25-29: Not clear: adaptable thermal properties of ice? configurable latent heat??? This should be constant, please elaborate. Authors answer: It means that the software user can define the thermal properties and adapt them to its specific settings (such has the melting point which is closer than -1°C than 0°C in high mountain bedrock), but they remain constant in the model.

7. p. 10, l. 6: Is this 5% value constant in the whole domain? Why not setting high porosity near the surface to somehow represent the presence of fractures and a smaller one in the interior of the wall? Authors answer: We agree that we could have considered a variable porosity throughout the model domain to pretend to setup more "realistic" parameters and better account for the variability of fracturing density. However, the fracturing density is basically unknown and according to our model evaluation the lack of consideration of porosity heterogeneity is not a major limit. However, we would recommend to better take this into consideration to simulate near surface processes and finer space and time scales. Instead of modifying the model setup without proper clues, we decided to address this comment by discussing this issue in a more detailed manner in section 5.1.4.

8. Figure 2: < -5 in colormap caption. Add (a) detailed MARST and (b) topographical situation. Authors answer: We wrote -5 instead of 5. However, none of the authors understood the following comment. MARST maps is given for the entire massif in Figure 1. Detailed MARST at the profiles location is given in Figure 3 together with the model geometry. Topographical details are in the text and well visible on this Figure 2.

9. Fig. 6: I would suggest to take another kind of marker (circle and bullet are too similar, and sometimes cannot be distinguished). Authors answer: Done, we replaced by triangles.  

Reviewer 2

1. However, from my point of view the work does not make a significant contribution neither to permafrost modelling nor to the understanding of high mountain permafrost.

Authors answer: As explained to Reviewer 1, we agree that the paper does not make progress in permafrost modelling or understanding. The aim of our paper is (i) to enhance recent developments in permafrost modelling by using the existing tools, procedures, and data, (ii) to show up the capabilities of simple modelling approach in reproducing rock wall thermal field, which, in existing studies, was not substantially demonstrated against field data, and (iii) enhance the latest climatic models driven by the most updated scenarios to (iv) propose quantitative and site-specific permafrost pathways for the future. Therefore the aim was not to provide new insight in permafrost studies but new insight in permafrost future changes, and we hoped that this topic would fit in "The evolution of permafrost in mountain regions" special issue.

2. The language is confusing throughout large parts of the manuscript. Further, the differentiation the authors seem to make between rock wall permafrost and high mountain permafrost in general - which is especially evident from the choice of the cited literature - is not convincing to me. Are there different hydro-thermal processes acting in rock walls than at less exposed sites or are they just active in different proportions? It is just stated that rock wall permafrost "is a relatively simple system" and it is further assumed to be homogeneous and saturated. The simplified assumption, that radiation and air temperature are sufficient for the simulation of such a system, especially if one of the relevant research questions in this context would be to gain insights on the triggering of future rock fall events (which are almost certainly not just a thermal phenomenon and would not occur in a homogeneous matter) - in my eyes - is to be considered as inadequate.

Authors answer: Well, in our opinion rock wall permafrost is significantly different than gentle mountain slope permafrost such as some modelling procedure are permitted to pretend realistic simulations of the thermal fields in rock walls but are not relevant for debris slopes strongly controlled by non conductive heat transfers. In the summarizing paper of the PACE project "Permafrost and climate in Europe : Monitoring and modelling thermal geomorphological and geotechnical responses" published by Harris et al. in Earth Sciences Review in 2009, a clear distinction is stated: "Progress has been made in modelling the major energy fluxes, but" ... "still not able to produce a sufficiently accurate estimation of snow cover and secondly, the coupling between atmosphere and ground where ground cover comprises coarse debris (non-conductive heat transfer) is not satisfactorily included. Therefore, as a first approach, more simple systems have been selected for detailed process modelling to avoid complex interactions with snow cover or complex materials such as coarse debris. Steep alpine rock walls represent such a system and allow more straightforward modelling with better estimation of uncertainties." Moreover, the evaluation step proposed in our paper clearly highlights the capabilities of such models to reproduce accurate thermal fields by only considering air temperature and solar radiation. Concerning the insight on rock fall triggering, we discussed this issue in a more detailed way in the section 5.3. of the revised version.

3. It is also stated, that only air temperature was used to drive the model, which is in contradiction to this already simplified assumption. The question also arises where the water is coming from, or going to, when freeze-thaw processes occur in the model?

Authors answer: Forcing data were based upon air temperature only, which is assumed to be the main driving factor of permafrost evolution in steep rock walls since rock wall permafrost studies have started. The system is saturated all the time, which means that the water doesn't go anywhere and doesn't need to be supplied.

4. I would encourage the authors to rewrite and restructure large parts of the manuscript and to address these questions as well as to refer to additional relevant literature on the subject. The uncertainties arising from the chosen simplifications should be discussed in this light and the conclusions drawn should also reflect that.

Authors answer: According to Reviewer 1 and external readers, the paper is clearly

written and it appears very difficult to rewrite and restructure large parts of the paper as suggested without knowing which parts are confusing. As Reviewer 1 suggested, we restructured Section 3 to present the model core and settings in a less confusing way. Uncertainty is largely discussed in Section 5 with 4 full pages dedicated to uncertainty discussion and encompassing all possible sources of uncertainty. This section was even reduced after preliminary review of the editor. We added 1 point to consider uncertainties in the reults in the conclusion (point 7. of the revised version).

Please also note the supplement to this comment:
http://www.the-cryosphere-discuss.net/tc-2016-132/tc-2016-132-AC1-supplement.zip

---

## Author Response (AR2)

Florence Magnin
Department of Geosciences
Sem Sælands vei 1
Geologibygningen
0371 Oslo

May, 16[th], 2017

Dear Christian Hauck and anonymous Reviewers,

We sincerely thank you for your relevant comments and questions that we tried to address in the best way in order to improve our paper quality.  The main changes that have been performed in the revised version are as follow:

- We submitted the paper to a native English speaker in order to correct the English grammar. We hope that the quality of the language is now sufficient for publication.
- We reduced the discussion by 1036 words (-36%), mainly by moving some paragraphs and section 5.3 in a perspectives section that comes after the conclusion and by removing some ideas that after revisions were not relevant any more or already expressed in other sections of the paper. In the perspectives section we also address some of the comments, such as the climate model resolution.

Here below you will find answers to your detailed questions and comments.

Concerning the Editor's comments, we only addressed those that were requiring additional explanations on top the changes in the text. Otherwise, all other comments have been directly addressed in the revised text.

Florence Magnin,

On behalf of all the co-authors

**Answers to editor:**

**Answer to comment P4 L24-25:** *the discussion about the heat transfer processes based on the temperature-depth profiles and the relevant literature (e.g. Williams and Smith, 1989) is detailed in Magnin et al., 2015b. Here we don't provide all these details again since this is not the purpose of our paper. However, we moved the reference to Magnin et al., 2015b at the end of the sentence to encourage the reader to have a look to the paper if he wants to know more about the Aiguille du Midi thermal regime.*

**P5 L9-10.** *When we submitted the paper we only had 1 ERT measurement for this site. Then, during fall 2016 we conducted additional measurements in the framework of another project. These new measurements will be published later in an appropriate publication. Therefore, there are not ready to be published yet.*

**P5 L11.** *We assess that it is warm permafrost at this site with resistivity data because the temperature-resistivity relationship has been calibrated in a cold room on a local rock sample. Thus we can say that given the range of resistivity values, this is very likely warm permafrost.*

**P5 L24-25.** *This is true that snow has an effect on rock wall temperature and we also have evidences that it influences the Aiguille du Midi borehole data (discussed later in the paper). What we meant here at this stage of the paper is that for long term simulation the snow cover effect may be neglected, and we used the term "relatively simple". However, this may be confusing and this may also neglect the most recent studies, so we have re-formulated the sentence.*

**P6 L30-35**: *The heat capacity is not used here, it is the volumetric heat capacity obtained by multiplying the specific heat capacity by the density and then annotated as the product of these two terms. We specified this combined expression into brackets in the revised version.*

**P7 L10-15** p.7 : *the expression is now into brackets.*

**P7 L1:** *adimensional means without dimension.*

**Line 20 p.7 :** *$lambda\_s$ is in the both sides because on the left side it is a $lambda\_s$ value at any time and $lambda\_{s,0}$ is the initial reference value of $lambda\_s$. Idem for $rho\_c$ in the equation 4 of the revised version (former equation 5).*

**Line 24 p.7 :** *the expression of the volumetric heat capacity have been modified in this equation such as in equation 1 in order to avoid any confusion between rho (density) and ($rho\_c$) volumetric heat capacity.*

**P8-9, L32-33.** *Yes the map is supposed to represented current permafrost conditions (considering that the transient effect for the period 1961-1990 still influences the bedrock thermal regime at depth), with a slight overestimation of permafrost conditions since these transient effects may also not be strong enough to still have a major influence on the current thermal regime at depth. But concerning the surface temperature, this is rather representative of the period 1961-1990, since MAAT from this period has been used to map MARST.  We added a few words to make the sentence clearer.*

**P 16, L12-13:** *This is true that  usually, that are the permafrost distribution models that use the km-scale grids data to map permafrost. However, some transient rock wall permafrost models also use*

*such data to force the simulation (e.g. Myhra et al., 2015 in PPP: Model distribution and temporal evolution of permafrost in steep rock walls along a latitudinal transect in Norway by CryoGRID 2D). To adjust our text to this comment we removed "commonly used", since this is not so common to use such approach. Also, we adapted the references in the following text in order to only refer to studies performing transient modelling (we remove references to Gruber, 2012 and Gisnås et al., 2013).*

**P16, L16-17.** *Here we talk about climate models data the SeNorge data in Norway for example or the SAFRAN model from Meteo France. We clarified the sentence accordingly*

**P21, Sect. 5.2.2.** *Here we think that even given the uncertainty in our study, this section is still worth, mainly because the key-message of the discussion on uncertainty is that short-term and shallow layer processes are not reliable, but that long-term and summit scale processes are reliable (and this is proven by the model evaluation). Also, we think that this is important for the reader to have such discussion on how to interpret the results and about what can be kept as main result once the uncertainty is taken into account.*

*However, we agree with the fact that shortening the discussion would improve it. Thu, we shortened it by moving section 5.3 to the perspectives section as you suggested. For this purpose, we built up a new section after the conclusions that defines our research perspectives. We also shortened the discussion by removing some sentences of paragraph that were either repeated in elsewhere in the manuscript, or that after all, were not of primary importance to understand the key-messages of the paper.*

**Answers to reviewer 3**

**1. p2 l4: what is 'this warning study' you mention?**

*This relates to the study from Haeberli et al., 1997 of the previous sentence. But since it was confusing, we reworked the sentences.*

**2. p2: reference to Blunden and Arnt is disingenuous as that publication refers to north slope boreholes whereas you are discussing steep rock boreholes in the Alps where to my knowledge trends are less clear (partly due to generally short measurement periods)**

*Here we refer to figure 2.7 pS16 of the referred report. This figures shows the trend of several European boreholes along a latitudinal transect which includes several boreholes in the European Alps drilled in various aspects, and which are also those having the longest time series. We still think that this reference fits in this paragraph shortly describing the history of mountain and rock wall permafrost monitoring in Europe so wekept it has it was.*

**p.20l6-7: you state that summer precip will 'certainly decrease' in the Alps. The given citation describes the study of Rajczak et al 2013 which shows that "that the frequency of wet days (left column) is projected to substantially decrease in summer across the entire Alpine region". I think you should rephrase your statement more carefully as**

**there is never certainty in model results. Perhaps: "Climate studies have shown a high likelihood that summer precipitation will decrease in the European Alps."**

*We agree with this comment. But to shorten a little bit the discussion, we decided to remove these few sentences and only explain why we didn't used precipitation scenarios.*

**3. As mentioned by previous reviewers, I still find grammar not quite up to scratch in numerous places and would recommend a second proof-read prior to publication.**

*A native English speaker has reviewed and corrected the manuscript before this last submission, which was not the case of our previous manuscripts. We hope the grammar is now correct enough for publication.*

**4. I am confused by the author's continued use of the word debatable, eg: "short time scale changes are debatable but beyond the scope of this study"**

*We don't use this word any more in our revised version.*

**5. I found it hard to trace the various comments/responses, I would suggest, in future, that the authors organise their responses clearly by each referee report in order to make this task easier.**

*It was actually organised by reviewers report but this was not clearly stated. We basically forget to state to write subtitles to describe to which reviewer we were answering. We perfectly understand that this was confusing and we apologize. In the future we will make sure that we don't forget this important detail any more. Thank you for noticing!*

[revised manuscript text omitted]

---

## Author Response (AR3)

Florence Magnin                                                                June, 21st, 2017
Department of Geosciences
Sem Sælands vei 1
Geologibygningen
0371 Oslo

Dear Christian Hauck,

Thank you very much for your careful reading and noticing all these technical issues. We performed all the required changes.

Concerning the 2 comments related to "multi-hm" and "hm-scale", which mean "multiple hectometers" and "a hectometric scale", we asked 2 native English speakers and they both agreed on the fact that it is correct and understandable. However, to avoid confusion we reformulated these expressions based on their suggestions.

We don't have any further comments since all the other required changes were making sense.

We hope that our paper now satisfies the standards for being published in TC.

Florence Magnin,

On behalf of all the co-authors